# A Phase II trial of alternating osimertinib and gefitinib therapy in advanced *EGFR*-T790M positive non-small cell lung cancer: OSCILLATE

Lavinia Tan[1,2], Chris Brown [3], Antony Mersiades[3], Chee Khoon Lee[3,4], Thomas John [1,2], Steven Kao[5], Genni Newnham[6], Kenneth O'Byrne[7], Sagun Parakh[8], Victoria Bray[9], Kevin Jasas[10], Sonia Yip[3], Stephen Q. Wong[1,2], Sarah Ftouni[1], Jerick Guinto[1], Sushma Chandrashekar[1], Stephen Clarke[11,12], Nick Pavlakis[11,12], Martin R. Stockler[3], Sarah-Jane Dawson [1,2,13,14] & Benjamin J. Solomon [1,2,14]

In this phase II, single arm trial (ACTRN12617000720314), we investigate if alternating osimertinib and gefitinib would delay the development of resistance to osimertinib in advanced, non-small cell lung cancer (NSCLC) with the epidermal growth factor receptor (*EGFR*) T790M mutation ($n = 47$) by modulating selective pressure on resistant clones. The primary endpoint is progression free-survival (PFS) rate at 12 months, and secondary endpoints include: feasibility of alternating therapy, overall response rate (ORR), overall survival (OS), and safety. The 12-month PFS rate is 38% (95% CI 27.5–55), not meeting the pre-specified primary endpoint. Serial circulating tumor DNA (ctDNA) analysis reveals decrease and clearance of the original activating *EGFR* and *EGFR*-T790M mutations which are prognostic of clinical outcomes. In 73% of participants, loss of T790M ctDNA is observed at progression and no participants have evidence of the *EGFR* C797S resistance mutation following the alternating regimen. These findings highlight the challenges of treatment strategies designed to modulate clonal evolution and the clinical importance of resistance mechanisms beyond suppression of selected genetic mutations in driving therapeutic escape to highly potent targeted therapies.

Intratumour genetic heterogeneity driving acquired resistance to targeted therapies remains a major impediment to improving clinical outcomes in cancer. Since the initial discovery of gatekeeper mutations, which render resistance to potent kinase inhibitors, a key unanswered question in cancer management has related to whether modulating selection pressures driving genetic evolution within tumors would alter the natural history of the disease and impact on clinical outcomes.

Activating mutations of the epidermal growth factor receptor genes (*EGFR*m) are key oncogenic drivers in non-small cell lung cancer (NSCLC), with 85% of cases arising from in-frame deletions of exon 19 (del19) or exon 21 L858R point substitution[1,2]. *EGFR*m confer sensitivity to treatment with EGFR tyrosine kinase inhibitors (TKIs), with evidence that first-generation (erlotinib and gefitinib) and second-generation (afatinib and dacomitinib) EGFR TKIs have significantly improved progression-free survival (PFS) in treatment naïve, advanced *EGFR*

A full list of affiliations appears at the end of the paper. e-mail: Sarah-Jane.dawson@petermac.org; ben.solomon@petermac.org

positive NSCLC when compared with chemotherapy[3–7]. Despite the initial benefit of TKIs, resistance inevitably occurs through various mechanisms. The most common route of therapeutic escape to first or second-generation EGFR TKIs is through the emergence of the *EGFR* T790M mutation in approximately 50–60% of cases[8]. These findings underpinned the development of osimertinib, a third-generation, irreversible, oral EGFR TKI that selectively inhibits both *EGFR* sensitizing and T790M resistance mutations by covalent binding to the C797 residue in the ATP-binding site of mutant EGFR[9,10] In patients with *EGFR* T790M mutation positive NSCLC who progressed on earlier generation TKIs, osimertinib is associated with superior response rates (71% *vs* 31%) and improved PFS (median 10.1 vs 4.4 months) compared with platinum-doublet chemotherapy[11]. Furthermore, osimertinib has become the standard first-line treatment based on improved OS (median 38.6 versus 31.8 months) and improved control of central nervous system metastases when compared with first-generation TKIs[12–14]. Nevertheless, acquired resistance to osimertinib remains a major therapeutic challenge, and novel treatment strategies are needed to prevent or delay the emergence of resistance in this setting.

To date, reports of resistance to osimertinib have largely focused on patients treated in the second or later line setting, with fewer studies characterizing resistance in the first-line setting[15–18]. In the AURA3 trial, the *EGFR* C797S resistance mutation within the kinase binding domain was detected in 15% of patients at progression on second-line osimertinib[17]. In addition, 50% of patients showed loss of *EGFR* T790M and this was associated with early resistance to osimertinib. Furthermore, loss of *EGFR* T790M was also associated with the presence of off-target resistance mechanisms, including emergence of secondary driver oncogene mutations or fusions, and histologic transformation[16,19,20]. Traditionally sequencing of tumor tissue from progressing lesions has been the cornerstone of understanding resistance mechanisms, however, more recently serial ctDNA analysis has proven to be an important complementary tool for monitoring and characterizing patterns of genomic evolution and resistance to TKI therapy[17].

Preclinical data has shown that the *EGFR* C797S mutation in the absence of T790M is sufficient to cause resistance to third generation TKIs whilst in some instances allowing the tumor to retain sensitivity to gefitinib[21]. This, together with the likely presence of non-T790M driven clones in the context of tumor heterogeneity, raises the possibility that patients with acquired resistance to osimertinib through this mechanism may benefit from combination treatment of osimertinib with an earlier generation TKI. Furthermore, the selection of *EGFR* T790M positive clones under therapeutic pressure from a first or second-generation EGFR TKI may provide an advantage by maintaining a dynamic equilibrium to prevent emergence of alternative mechanisms of resistance that are less amenable to targeted therapy. Based on this hypothesis, we designed and conducted OSCILLATE, a single arm, phase 2 trial, to test the hypothesis that a temporally defined combination of alternating osimertinib and gefitinib would alter selection pressures that drive clonal evolution to delay the development of resistance to osimertinib. The study evaluated the efficacy and safety of this treatment strategy in patients with advanced, *EGFR* T790M positive NSCLC. Serial ctDNA analyses were performed to assess therapeutic response, understand clonal dynamics, and characterize genomic mechanisms of resistance to alternating therapy. Here, we report the results.

## Results
### Patient characteristics
Forty-seven participants with *EGFR* T790M mutation positive NSCLC with disease progression after first generation EGFR TKIs were enrolled between September 4, 2017 and June 11, 2019 and treated with induction osimertinib followed by an alternating regimen of gefitinib and osimertinib (Fig. 1a, b). Two additional participants

were enrolled but found to be ineligible for study treatment and did not receive study drug and were excluded from all safety and efficacy analyses: one had symptomatic brain metastases, the other had deranged liver function. Demographics and baseline characteristics of the 47 participants are summarized in Table 1. The median age was 60 years (range 32–86), 62% were female, 66% were never smokers, 60% were non-Asian, and the median number of prior systemic therapies was 1 (range, 1–2). The most common activating *EGFR* mutation based on tumor genotyping was exon 19 deletion (64%), followed by exon 21 L858R (34%) and all patients had *EGFR* T790M confirmed through tumor (45%) and/or ctDNA analysis (55%), based on local laboratory testing at each participating site.

### Efficacy and feasibility
By the time of data cut-off, with a median follow up of 37 months, 41 of 47 participants (87%) had experienced disease progression. Two participants experienced disease progression after 8 weeks of induction osimertinib and did not proceed with alternating therapy. The primary endpoint of the study, the proportion progression free at 12 months was 38% (95% CI 27–55), below the pre-specified target of 23/41 progression free at 12 months (Fig. 2a). Secondary outcomes including median PFS was 9.4 months (95% CI 7.2–13.0) (Fig. 2a), and the median time from first progression (PFS1) to second progression (PFS2) on continuous osimertinib after stopping alternating therapy was 3.8 months (95% CI 3.4–5.9) (Supplementary Fig. 1). The median OS was 26 months (95% CI 19-not estimable (NE)) (Fig. 2b).

Tumor response assessed according to the Response Evaluation Criteria in Solid Tumors (RECIST) v1.1 is shown in Fig. 2c–e and Table 2. The confirmed objective response rate (ORR) was 45% (95% CI 31–59%) with 21/47 evaluable participants demonstrating a partial response (PR). Stable disease (SD) was observed in a further 17 participants (36%). The disease control rate (CR + PR + SD) was 81%. The median duration of response was 9 months (95% CI 7-NE). Progressive disease (PD) without prior CR, PR, or SD, was observed in 9 participants (19%), in 2 of which PD had occurred before starting alternating therapy (Table 2).

Unplanned subset analysis did not show a difference in PFS between those with an *EGFR* exon 19 deletion compared to with exon 21 L858R (HR for PFS 1.8, 95% CI 0.93–3.40, *P* = 0.08) (Fig. 2f). There was also no difference in OS observed in participants with an exon 19 deletion versus exon 21 L858R mutation (HR for OS 1.8, 95% CI 0.85–4.0, *P* = 0.12) (Supplementary Fig. 2). Unplanned analysis of the primary endpoint by sex did not identify any differential effect, with similar participants progression free at 12 months between females 10/29 (34%) and males 5/18 (28%). In addition, no difference was observed in PFS (HR 1.28, 95% CI 0.68–2.40, *P* = 0.4) and OS (HR 1.11, 95% CI 0.52–2.38, *P* = 0.8) in female versus male participants (Supplementary Fig 3).

Sixty-eight percent of participants (32/47, 95% CI 54–80) completed alternating therapy without any dose interruption for 6 months, with a median of 9.4 months on therapy. Thirty-three participants switched from alternating therapy to continuous osimertinib following disease progression. Reasons for discontinuing all trial treatment were disease progression (60%), end of study (26%), adverse event (AE) (4%), clinician preference (4%), patient preference (4%), and death (2%). There were 28 deaths observed during the trial follow-up: 27 due to progressive disease and 1 due to COVID-19 respiratory failure in the absence of progressive disease.

### Safety
The most frequent AEs of any grade were diarrhoea (40%), fatigue (38%), acneiform rash (38%), cough (36%), and headache (36%) (Table 3). There were no reported cases of drug-related interstitial lung disease. There were 2 participants who discontinued study treatment

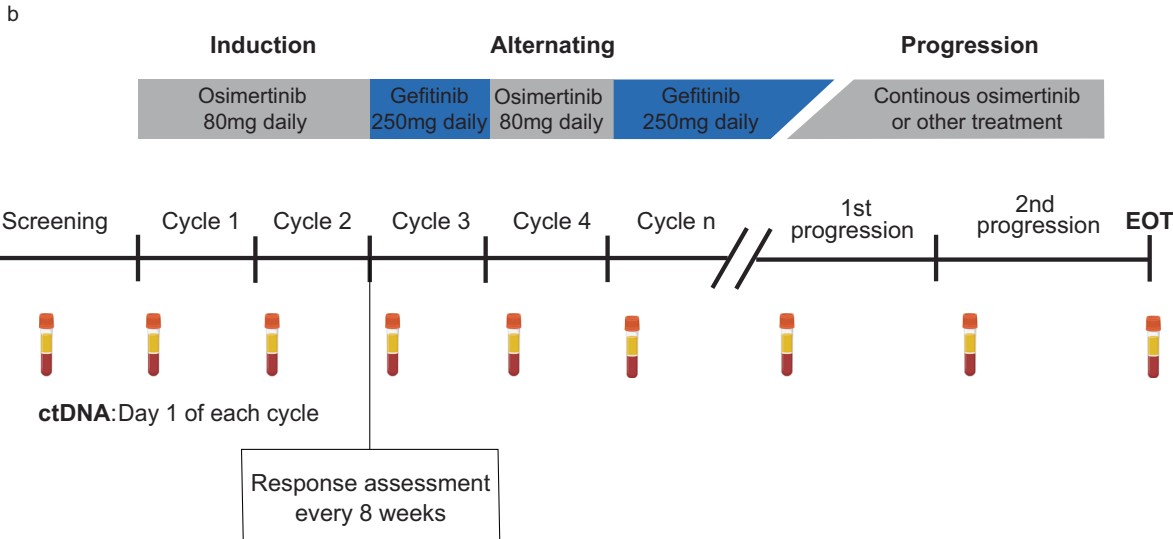

**Fig. 1 | Design of the OSCILLATE trial. a** Clinical trial consort diagram with final number of participants recruited and analyzed. Plasma samples were collected serially as shown. Baseline and progression plasma samples were characterized using the assays listed for the detection of potential genomic resistance markers. **b** Clinical trial design of the OSCILLATE study and overview of plasma sample collection for translational research. Participants with *EGFR* T790M positive advanced NSCLC were treated with an alternating regimen of osimertinib and gefitinib until progression, unacceptable toxicity, or death. NSCLC non-small cell lung cancer. ctDNA circulating tumor DNA.

(alternating therapy or continuous osimertinib) due to AEs and there were no treatment-related deaths.

## Baseline ctDNA analysis prior to treatment

Baseline pre-treatment plasma DNA detected *EGFR*m and T790M in 78% (36/46) and 76% (35/46) of participants, respectively (Supplementary Table 1). The most frequent *EGFR*m detected in baseline plasma were exon 19 deletion (23/36, 64%), exon 21 L858R (12/36, 33%), and exon 21 L816Q (1/36, 3%) (Fig. 3a, b, and Supplementary Table 1). Seventy percent (32/46) of participants had detectable ctDNA levels of both *EGFR*m and T790M in their baseline plasma DNA, while 15% (7/46) had neither detectable *EGFR*m nor T790M and were considered non-shedders (Fig. 3a). In line with previous reports[22], non-shedders had smaller baseline tumor target lesion sizes than shedders (median sum of longest diameter, 32 mm [range, 15–58 mm] versus 48 mm [range, 15–140 mm], $P = 0.09$).

Baseline plasma DNA analysis identified at least 1 co-occurring mutation in ctDNA of 39/46 (85%) participants (Fig. 3b and Supplementary Data 1), with an average of 3 mutations per participant (range 0–7). The most common co-occurring mutations identified included alterations in *TP53* (59%), *CTNNB1* (18%), and *MET* (8%). We observed no differences in PFS and OS between participants without and with a *TP53* mutation (HR for PFS 1.1, 95% CI 0.54–2.09, $P = 0.85$, HR for OS 0.81, 95% CI 0.37–1.81, $P = 0.62$, Supplementary Fig. 4). Copy number alterations with amplifications of *EGFR*, *MET* and *ERRB2* were also identified in 19 participants (50%) (Fig. 3b). The presence of *EGFR* amplification was associated with shorter PFS (median 5 versus 11 months), compared to those without *EGFR* amplification (HR 2.2, 95% CI 1.08–4.43, $P = 0.01$) (Supplementary Fig. 5a), and these participants also showed a trend towards shorter OS (median 15 versus 28 months; HR 2.0, 95% CI 0.88–4.64, $P = 0.08$) (Supplementary Fig. 5b). Similarly, *MET* amplification was associated with shorter PFS (median PFS 4 versus 9 months, HR 1.7, 95% CI 0.74–4.07, $P = 0.13$, Supplementary Fig. S6a), and OS (median OS 15 versus 26 months, HR 1.6, 95% CI 0.60–4.41, $P = 0.28$, Supplementary Fig. 6b).

Quantitative analysis of the level of ctDNA (copies/ml of plasma) for both the *EGFR*m and T790M showed that at baseline, levels were significantly higher among those with PD as their best response (Fig. 3c, d). We next assessed whether the ratio of T790M to *EGFR*m (T790M/EGFRm$_R$) at baseline was associated with treatment response. We defined the T790M/EGFRm$_R$ in baseline plasma as the ratio of the T790M mutation (copies/mL) relative to the *EGFR*m (copies/mL), and used it as surrogate measure of the abundance of T790M-positive clones. Among 31 participants with both the activating *EGFR*m and T790M detectable in baseline plasma, the median T790M/EGFRm$_R$ prior to starting treatment was 0.21 (range, 0.02–0.83). Although not statistically significant, the median ratio was higher among those who achieved a PR compared to those with PD as their best response (median ratio: 0.32 vs 0.17, $P = 0.10$). Those with a ratio below the median, compared with above the median, had shorter PFS (median 4 versus 13 months; HR 2.8, 95% CI 1.24–6.40, $P = 0.006$) and OS (median 17 months versus NR; HR 3.4, 95% CI 1.26–9.42, $P = 0.01$) (Fig. 3e, f). The most frequent co-occurring alterations among participants with a lower than median T790M/EGFRm$_R$ were *TP53* mutations (80%, $P = 0.07$), *EGFR* amplification (67%, $P = 0.16$), and *MET* amplification (40%, $P = 0.72$) however, differences in the frequency of these co-alterations according to whether the T790M/EGFRm$_R$ was above or below the median, were not significant (Supplementary Fig. 7). We conducted a post-hoc analysis according to baseline *MET*-amplification. Within 36 non-*MET*-amplified patients, RR was higher 18/36 (56%) versus 2/11 (18%), PFS was longer 11 versus 3.7 months (HR 2.17, 95% CI 1.06–4.55) and OS was longer 38 versus 15 months (HR 1.82, 95% CI 0.76–4.35).

**Table 1 | Participant demographics and baseline characteristics**

| Characteristics | *n* (%) |
|---|---|
| Median age (range) | 60 years (32–86) |
| **Sex** | |
| Female | 29 (62) |
| Male | 18 (38) |
| **ECOG performance status** | |
| 0 | 23 (49) |
| 1 | 22 (47) |
| 2 | 2 (4) |
| **Smoking status** | |
| Current smoker | 1 (2) |
| Former smoker | 15 (32) |
| Never smoker | 31 (66) |
| **Ethnicity** | |
| Asian | 19 (40) |
| Non-Asian | 28 (60) |
| ***EGFR* driver mutation** | |
| Exon 19 deletion | 30 (64) |
| Exon 21 L858R substitution | 16 (34) |
| Exon 21 L861Q mutation | 1 (2) |
| Exon 20 S768I mutation | 1 (2) |
| **Baseline T790M detection** | |
| Plasma | 26 (55) |
| Tissue | 21 (45) |
| **Sites of metastatic disease** | |
| Brain | 14 (21) |
| Leptomeningeal | 1 (2) |
| Liver | 9 (19) |
| **No of prior systemic therapy** | |
| 1 | 44 (94) |
| 2 | 3 (6) |

## ctDNA dynamics following induction and alternating therapy

Forty-six participants had serial baseline, week 4 (following induction therapy), and week 12 (following first alternating therapy) plasma samples collected (Fig. 1a) facilitating analysis of ctDNA a key correlative endpoint of the study. A total of 36 paired baseline-week 4 (BL-W4) and 34 paired baseline-week 12 (BL-W12) plasma samples were analyzed for early EGFRm ctDNA dynamics (Supplementary Fig 8). A significant reduction in EGFRm levels (copies/mL) was seen between baseline to week 4 (BL-W4) and baseline to week 12 (BL-W12) in participants who achieved either a PR or SD, with the depth of response greatest in those with PR (Fig. 4a). In contrast, for participants with PD, whilst a significant decrease was initially seen in the BL-W4 EGFRm levels following osimertinib induction therapy, by week 12 following alternating gefitinib treatment, the EGFRm levels were steadily increasing and returning to close to pre-treatment levels (Fig. 4a). The circulating DNA ratio (CDR)[23] was used to measure the mutation abundance (copies/mL) at week 4 and week 12 relative to baseline. Most participants (week 4: 36/36; 100% and week 12: 31/34; 91%) had a week 4-baseline ratio (W4-BL$_R$) and a week 12-baseline ratio (W12-BL$_R$) of <1, with a median of 0.00 and 0.03 respectively, indicating a marked decrease in ctDNA.

We next assessed early T790M mutation dynamics through ctDNA analysis and compared these changes to the EGFRm dynamics described above. A total of 36 paired baseline-week 4 (BL-W4) and 34 paired baseline-week 12 (BL-W12) plasma samples were analyzed (Supplementary Fig 8). In contrast to EGFRm, all participants showed a

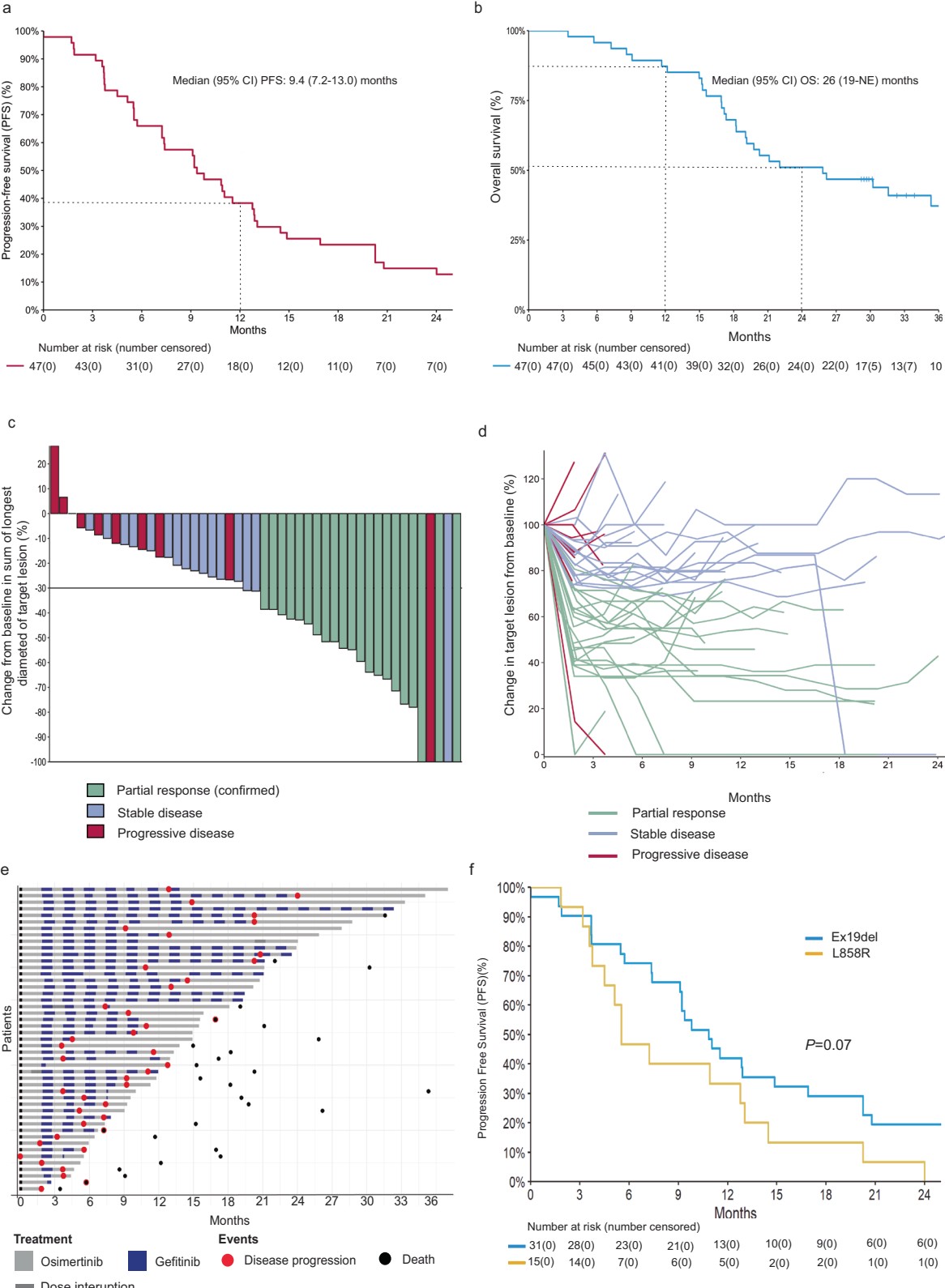

**Fig. 2 | Efficacy of alternating osimertinib and gefitinib in participants with *EGFR*-T790M positive advanced NSCLC. Source data are provided as source data file. a** Kaplan-Meier estimate of progression-free survival (PFS) for evaluable participants (*n* = 47). **b** Kaplan-Meier estimate of overall survival (OS) for evaluable participants (*n* = 47). **c** Waterfall plot with best percentage tumor change from baseline target lesions and best confirmed overall response for evaluable participants (*n* = 47). **d** Spider plot demonstrating response to treatment for each participant over time (*n* = 47). **e** Swimmer plot showing duration on treatment for 47 evaluable participants. Individual participants represented as lines. Line segments are colored according to treatment, with blue representing gefitinib and grey representing osimertinib, and dark grey representing dose interruptions. A red dot indicates progressive disease, and a black dot indicates death. **f** Kaplan-Meier estimate of PFS for evaluable participants (*n* = 46) stratified according to activating epidermal growth factor receptor (*EGFR*) mutation (EGFRm) type. Ex19del, exon 19 deletion, 95% CI, 95% confidence interval.

**Table 2 | Summary of efficacy data**

| Assessment | n = 47 |
|---|---|
| Best overall response, n (%) | |
| Partial response (PR) | 21 (45) |
| Stable disease (SD) | 17 (36) |
| Progressive disease (PD) | 9 (19) |
| Confirmed ORR (%) | 21 (45) |
| Median duration of response, months | 9.2 (7.4, NE) |
| Disease control rate (PR + SD) (%) | 38 (81) |
| Median PFS1#, months | 9.4 (7.2, 13) |
| Median PFS2*, months | 3.8 (3.4, 5.9) |

#PFS1 was defined as the interval from registration to the first occasion at which progression or death occurred.

*PFS2 was defined as time to progression or death on continuous dosing osimertinib, after alternating therapy was ceased.

*PFS* progression free-survival, *ORR* overall response rate.

**Table 3 | Summary of any grade treatment-related adverse events reported in at least 10% of participants**

| Adverse events | All participants (n = 47) n (%) | |
|---|---|---|
| | All grades | Grades ≥3 |
| Diarrhoea | 19 (40) | 0 (0) |
| Fatigue | 18 (38) | 1 (2) |
| Acneiform rash | 18 (38) | 0 (0) |
| Cough | 17 (36) | 0 (0) |
| Headache | 17 (36) | 2 (4) |
| Back pain | 16 (34) | 1 (2) |
| Musculoskeletal and connective tissue disorder | 15 (32) | 1 (2) |
| Nausea | 14 (30) | 2 (4) |
| Pain | 14 (30) | 1 (2) |
| Skin and subcutaneous tissue disorders | 13 (28) | 0 (0) |

significant decline in T790M levels at week 4 and week 12 compared to baseline regardless of response achieved (Supplementary Fig. 9). In keeping with these findings, the W4-BL$_R$ and W12-BL$_R$ of T790M levels was <1 in 94% (34/36) and 97% (33/34) of participants respectively.

We explored whether early, dynamic changes in ctDNA levels were prognostic of outcomes to alternating therapy. We observed that both decreased EGFRm W4-BL$_R$ and W12-BL$_R$ were associated with objective response (Supplementary Fig. 10a and Supplementary Fig. 10b). Conversely, the T790M W4-BL$_R$ and W12-BL$_R$ were not associated with response (Supplementary Fig. 11a and b).

We next investigated whether W4-BL$_R$ and W12-BL$_R$ were prognostic of PFS and OS and performed landmark analyses at week 4 and 12. Participants who had disease progression and/or death at those timepoints were removed from all analyses. Participants were also excluded due to lack of detectable ctDNA at baseline, week 4, and week 12 (Supplementary Fig. 8). Participants with an EGFRm W4-BL$_R$ above the median had shorter PFS (median PFS 5 versus 12 months, HR 2.6, 95% CI 1.2–5.3, P = 0.013) (Supplementary Fig. 12a). Likewise, OS was shorter in participants with a higher than median W4-BL$_R$ (median OS 16 months versus NR; HR 3.0, 95% CI 1.19–7.62, P = 0.008) (Supplementary Fig. 12b). Comparatively, although the T790M W4-BL$_R$ was not prognostic of PFS, OS was shorter in participants with a higher than median W4-BL$_R$ (HR for PFS 1.2, 95% CI 0.6–2.8, P = 0.6 and HR for OS 2.9, 95% CI 1.2–7.4, P = 0.021) (Supplementary Fig. 13a and b).

In contrast to the week 4 timepoint, ctDNA dynamics at the week 12 timepoint were a stronger prognostic factor of outcome. Participants with an EGFRm W12-BL$_R$ above the median demonstrated both an inferior PFS and OS (median PFS 3 versus 10 months; HR 2.4, 95% CI 1.04–5.58, P = 0.02 and median OS 15 months versus NR; HR 4.2, 95% CI 1.69–10.6, P = 0.002) (Supplementary Fig. 14 and Fig. 4b). Similarly, participants with T790M W12-BL$_R$ above the median had inferior PFS and OS (median PFS 4 versus 10 months; HR 4.1, 95% CI 1.7–9.7, P = 0.001 and median OS 13 months versus NR; HR 4.2, 95% CI 1.49–11.9, P = 0.001) (Supplementary Fig. 15 and Fig. 4c).

In addition to assessing the reduction in ctDNA abundance, we evaluated the use of ctDNA clearance as a potential surrogate marker of treatment response. We assessed clearance of ctDNA at week 4 and week 12 in participants with available plasma at those timepoints (Supplementary Fig. 8), with clearance defined as lack of detectable ctDNA compared to baseline. Clearance of EGFRm ctDNA occurred in 53% (19/36) and 38% (13/34) of participants at 4 weeks and 12 weeks, respectively. Whilst participants with clearance of EGFRm at either week 4 or week 12 experienced a longer PFS and OS compared to participants without clearance, undetectable EGFRm ctDNA at the week 12 timepoint was the strongest prognostic factor of PFS and OS

(median PFS 12 versus 5 months; HR 3.1, 95% CI 1.4–7.2, P = 0.008 and median OS NR versus 15 months; HR 9.3, 95% CI 3.8–23, P = 0.0002) (Supplementary Figs. 16 and 17 and Fig. 4d). In comparison to EGFRm clearance, a higher number of participants (4 weeks: 27/35; 77% and 12 weeks: 20/33; 61%) had clearance of T790M at 4 weeks and 12 weeks respectively. In keeping with the EGFRm findings, clearance of T790M at the week 12 timepoint was the strongest prognostic factor of PFS and OS (median PFS 10 versus 3 months; HR 4.5, 95% CI 1.9–11, P < 0.001 and median OS NR versus 9 months; HR 3.9, 95% CI 1.5–11, P = 0.006) (Supplementary Figs. 18 and 19 and Fig. 4e).

## Serial ctDNA analysis to evaluate genomic evolution

Plasma DNA collected at the time of disease progression was analyzed through targeted sequencing to identify evidence of genomic evolution following treatment and potential mechanisms of acquired resistance to therapy. In total, 27 participants who completed alternating therapy had plasma DNA available at disease progression, allowing comparison to the baseline pre-treatment ctDNA analysis (Fig. 4f and Supplementary Data 1). Notably, there was no evidence of EGFR C797S detected in progression plasma in any participant after alternating therapy. Two (7%) participants maintained T790M, while loss of T790M was seen in 70% (19/27) of participants at the time of progression. Of the 19 participants with T790M loss, 3 (16%) participants developed EGFR-dependent mechanisms of resistance with an acquired EGFR R451C mutation (n = 1) and EGFR amplification (n = 2). The other 6 (32%) participants developed EGFR-independent resistance alterations which included acquired mutations in PIK3CA, KRAS, MET, and PIK3R1. More than half of the participants (10/19, 53%) with T790M loss had no other genomic resistance mechanisms identified.

We next analyzed progression plasma samples in 19 participants who proceeded with continuous osimertinib following progression on alternating therapy and compared this to baseline pre-treatment ctDNA analysis (Fig. 4f and Supplementary Data 1). Of note, participants OSC8 and OSC17 progressed on induction osimertinib and were also included in this analysis. In contrast to our findings following alternating therapy, emergence of the EGFR C797S mutation was detected in 10% (2/21) of participants. Both participants (OSC014 and OSC029) harboured an EGFR exon 19 deletion mutation, however, only OSC014 had detectable T790M on plasma testing at baseline. EGFR amplification and a SMAD4 mutation were detected in the baseline plasma of OSC014, however, there were no detectable mutations in the baseline plasma of OSC029. Progression plasma of OSC014 post alternating therapy and continuous osimertinib revealed acquired MET amplification and a TP53 mutation in addition to the EGFR C797S mutation, with T790M mutation maintained at both timepoints.

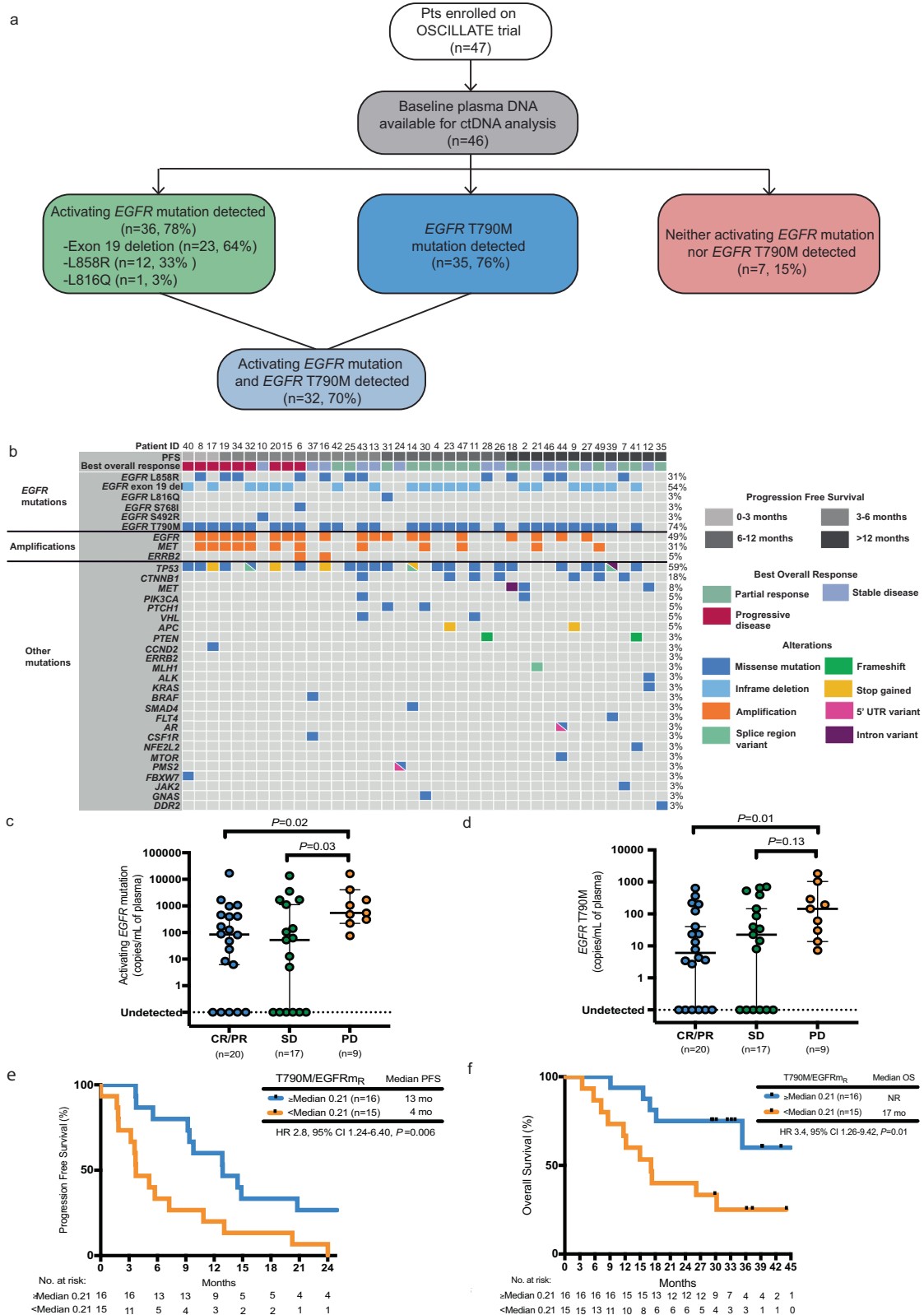

OSC029 had a detectable *FGFR2* mutation post alternating therapy and no other detectable genomic mutation post continuous osimertinib, apart from the *EGFR* C797S mutation.

Overall, two participants (10%) maintained T790M, and 15 participants (71%) had loss of T790M at the time of progression on continuous osimertinib. OSC11 had undetectable T790M at first

progression which became detectable again at the time of 2nd progression. T790M loss was observed in 2 participants (13%) after induction osimertinib and 1 participant (7%) at the time of 2nd progression. The remaining 12 participants (80%) had loss of T790M after alternating therapy and T790M remained undetectable at the time of 2nd progression. *EGFR* amplification was detected in 19% (4/21) of

**Fig. 3 | Baseline ctDNA analysis. Source data are provided as source data file.** **a** Consort diagram of plasma samples analyzed with droplet digital PCR and targeted capture-based sequencing assays. **b** Landscape of somatic mutations detected through targeted sequencing of baseline plasma DNA in 38 participants enrolled on the OSCILLATE trial. Each column represents an individual participant, and each row indicates a specific alteration. Participants are grouped by progression free-survival (indicated by grey shaded colored boxes) in the order of 0–3 months, 6–12 months, >6 months and >12 months. The bar below shows best radiological response assessment. The colour of bars is indicative of the type of mutation with grey = wild-type. Percentages listed right represents the proportion of participants harboring an alteration in the gene listed left. **c** Baseline EGFRm (copies/mL) in participants achieving PR, SD, or PD. *P* values represent PR vs. PD and SD vs. PD by two-sided Mann-Whitney test. Data are presented as median value

± 95% confidence interval. Each dot represents a single participant. **d** Baseline T790M mutation (copies/mL) in participants achieving PR, SD, or PD. *P* values represent PR vs. PD and SD vs. PD by two-sided Mann-Whitney test. Data are presented as median value ± 95% confidence interval. Each dot represents a single participant. **e** Kaplan-Meier estimate of PFS for participants ($n = 31$) stratified by median ratio of T790M to EGFRm (T790M/EGFRm$_R$) at baseline. Comparisons were made using a two-sided log-rank test and no adjustments were made for multiple comparisons. **f** Kaplan-Meier estimate of OS for participants ($n = 31$) stratified by median T790M/EGFRm$_R$ at baseline. Comparisons were made using a two-sided log-rank test and no adjustments were made for multiple comparisons. EGFRm epidermal growth factor receptor mutation, PR partial response, SD stable disease, PD progressive disease, PFS progression free-survival, OS overall survival.

participants and other potential resistance mechanisms included *TP53* mutations (5/21, 24%), *PIK3CA* mutations (5/21, 24%), and *MET* amplification (4/21, 19%).

## Discussion

The OSCILLATE trial evaluated the safety, efficacy and ctDNA correlates of a temporal combination of alternating osimertinib and gefitinib as second-line treatment of *EGFR*-T790M positive NSCLC. Alternating therapy was feasible and well tolerated, with toxicities consistent with the profiles of each agent. Despite a lower overall response rate of 45% compared to the AURA3 trial, the median PFS of alternating therapy (9.4 months) was comparable with the median PFS of continuous osimertinib (10.1 months) in the second-line setting[11]. However, the trial did not meet its prespecified 12-month PFS primary efficacy endpoint. Participants were allowed to continue on osimertinib monotherapy after progression on alternating therapy, with an additional median PFS2 of 3.8 months. Overall survival with the alternating regimen was 25.8 months, consistent with that reported for osimertinib monotherapy[11]. The characteristics of the participants in our study were similar to that of the AURA3 trial, with the exception of a higher proportion enrolled based on positive T790M detected through plasma testing only.

Our detection rate of activating *EGFRm and EGFR* T790M ctDNA in baseline pre-treatment plasma samples was in line with previous reports[24,25], supporting the use of plasma EGFRm and T790M testing when tumor biopsy is not possible. In the second line setting, osimertinib is recommended based on the detection of the T790M resistance mutation, however, various studies have shown that the abundance of T790M-positive clones can influence response to osimertinib[26,27]. Tumors with a lower fraction of T790M clones are associated with inferior responses to osimertinib likely due to the presence of other resistant clones. In our cohort of patients, the median ratio of T790M to EGFRm in the baseline plasma was 0.21, which is lower than previous studies[26,27]. Correspondingly, we demonstrated that participants with a low T790M to EGFRm ratio had shorter PFS and OS with alternating therapy.

Monitoring of ctDNA levels has been shown to be a useful tool for assessing tumor response to targeted therapy and early ctDNA dynamics have been shown to be prognostic of clinical outcomes in several studies[8,16]. Here, we analyzed ctDNA levels at baseline, 4, and 12 weeks allowing us to characterize the unique and contrasting clonal dynamics between single agent osimertinib and alternating osimertinib and gefitinib therapy. When examining response dynamics, both EGFRm and T790M mutation ctDNA levels showed a rapid decline by 4 weeks, indicative of the rapid response to continuous osimertinib in T790M mutant disease. However, the early decrease in EGFRm levels and clearance of EGFRm ctDNA were stronger prognostic factors of treatment response and clinical outcomes than changes in T790M following continuous osimertinib treatment. In contrast, when assessing ctDNA dynamics at 12 weeks, both EGFRm and T790M levels were important determinants of

outcome to alternating therapy. Individuals showing reduction and/or clearance of both EGFRm and T790M ctDNA demonstrated the longest PFS and OS, highlighting the ability of alternating therapy to continuously suppress both EGFRm and T790M clonal populations in some participants.

We characterized the baseline genomic landscape through pre-treatment ctDNA analysis and identified co-occurring mutations in *TP53* in up to 59% of participants. Although other studies and a recent meta-analysis have demonstrated worse prognosis for patients with both *EGFR* and *TP53* mutations[28,29], the presence of concurrent *TP53* mutations did not impact clinical outcomes of participants treated on alternating therapy in our trial. In another recent study by Chabon *et al.*, the presence of multiple resistance mechanisms following initial EGFR TKI therapy was associated with inferior response to third generation TKIs, with copy number gains in *MET* and *EGFR* more common in patients with intrinsic resistance[30]. Likewise, we demonstrated that the presence of *EGFR* and *MET* amplification at baseline was associated with shorter PFS, likely as a mechanism of primary resistance to alternating therapy.

Whilst patterns of genomic evolution in patients receiving monotherapy with first and third generation TKIs have been well characterized, there is currently no understanding of the impact of alternating therapy on the evolutionary trajectories of clonal resistance in the clinical context. Using ctDNA, we characterized evolving genomic changes throughout treatment and at the time of disease progression. The most apparent findings we observed at progression were a higher rate of T790M loss compared to ctDNA analyses from the AURA3 trial (70% versus 49%) as well as the absence of C797S mutations among participants treated with the alternating therapy[17]. These findings demonstrated that alternating therapy was effective at supressing these distinct *EGFR* mutant clones. Loss of T790M was associated with alternative resistance mechanisms such as bypass signalling pathway and EGFR-independent resistance, many of which have been previously reported[31]. However, a vast number of participants in our cohort had loss of T790M and no other genomic mechanism of resistance identified through ctDNA analysis. Given the limited availability of matched tumor tissue samples at progression, we were not able to identify other mechanisms of resistance such as histological small cell transformation[8,31]. Furthermore, our analysis was limited to the mutations covered by our targeted sequencing panel and other potential genetic and non-genetic resistance mechanisms were not able to be explored due to lack of tissue. Acquired *EGFR* C797S mutations were not detected in progression plasma after alternating therapy but emerged in 2 participants after osimertinib mono-therapy, raising the possibility that alternating therapy can delay the emergence of *EGFR* C797S mutations. However, given the small cohort size these findings remain exploratory and will require independent validation in a larger cohort. The identification of heterogenous resistance alterations in this study highlights the importance of repeat tumor or liquid biopsies in patients

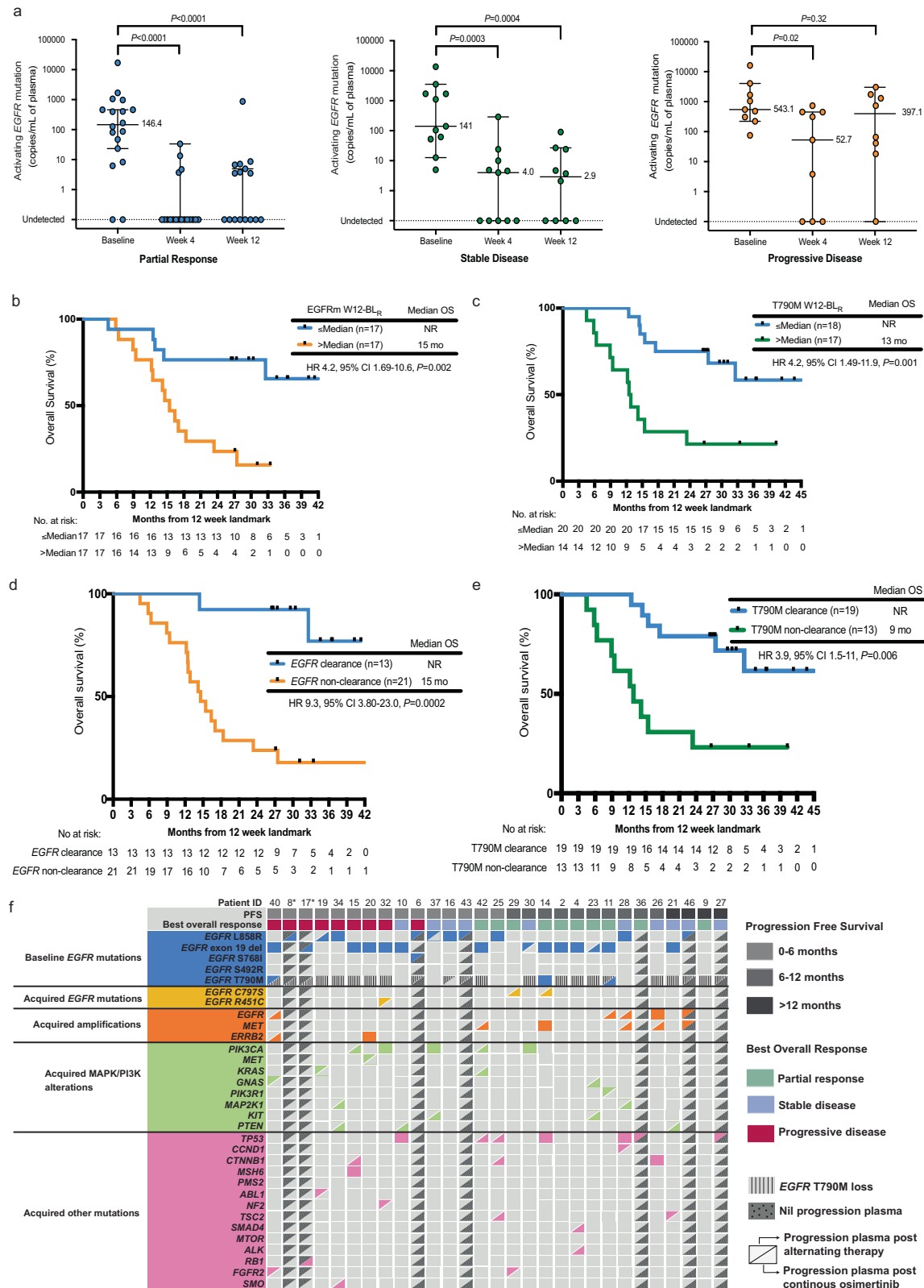

progressing on targeted therapies to inform future treatment strategies targeting clonal evolution.

We acknowledge several limitations of our study, particularly the small cohort of participants in a single arm study with no comparison arm of osimertinib monotherapy. Although our alternating strategy of osimertinib and gefitinib demonstrated comparable PFS to the AURA3 trial, osimertinib is now the preferred first-line treatment based on

improved survival benefit over first-generation TKIs. The efficacy of gefitinib may be limited in our study as participants had already progressed on a first-generation TKI. Furthermore, participants with baseline *MET* amplification were included in this study, with subset analysis revealing a lower response rate to alternating therapy. Current efforts are underway to test the combination of newer agents targeting *EGFR* mutations in NSCLC such as amivantamab, an EGFR-MET

**Fig. 4 | ctDNA dynamics and genomic alterations at disease progression.**
**Source data are provided as source data file. a** Dynamics of EGFRm (copies/mL) between baseline and week 4 (*n* = 36), and baseline and week 12 (*n* = 34) in participants achieving PR, SD, and PD. Data are presented as median ctDNA copies/mL ± 95% confidence interval. Comparisons between baseline and week 4, and baseline and week 12 were made using a two-tailed Wilcoxon signed-rank test. All *P* values are exact. Each dot represents a single participant. **b** Kaplan-Meier estimate of OS for participants (*n* = 34) stratified according to EGFRm W12-BL$_R$ ≤median and >median. Comparisons were made using a two-sided log-rank test and no adjustments were made for multiple comparisons. **c** Kaplan-Meier estimate of OS for participants (*n* = 35) stratified according to T790M W12-BL$_R$ ≤median and >median. Comparisons were made using a two-sided log-rank test and no adjustments were made for multiple comparisons. **d** Kaplan-Meier estimate of OS for participants (*n* = 34) stratified according to clearance versus non-clearance of EGFRm DNA (copies/mL) by week 12. Comparisons were made using a two-sided log-rank test

and no adjustments were made for multiple comparisons. **e** Kaplan-Meier estimate of OS for participants (*n* = 32) stratified according to clearance versus non-clearance of T790M mutant DNA (copies/mL) by week 12. Comparisons were made using a two-sided log-rank test and no adjustments were made for multiple comparisons. **f** Summary of ctDNA genomic features at disease progression in 27 and 21 participants who completed alternating therapy and proceeded with continuous osimertinib following progression on alternating therapy, respectively with an available progression plasma sample. Participants are ordered according to time to progression (indicated by grey shaded colored boxes) in the order of 0–6 months, 6–12 months, and >12 months. Each column represents an individual participant, and each row indicates a specific acquired alteration. The colour of bars is indicative of the type of mutation with grey = wild-type.*Participants 8 and 17 progressed on induction osimertinib and never proceeded to alternating therapy. EGFRm epidermal growth factor receptor mutation, PR partial response, SD, stable disease, PD progressive disease, W12-BL$_R$, week 12 to baseline ratio, OS overall survival.

bispecific antibody[32], and patritumab deruxtecan (HER3-DXd), an antibody-drug conjugate[33], with osimertinib, as they are able to target a diverse range of resistance mechanisms.

In conclusion, OSCILLATE demonstrated that for patients with T790M acquired resistance, alternating osimertinib and gefitinib was a therapeutic strategy with similar activity to continuous osimertinib. Alternating therapy was associated with distinct clonal dynamics observed through serial ctDNA analysis with successful suppression of both EGFRm and T790M clonal populations, and delayed emergence of C797S mediated resistance in some patients. Our findings provide important insights to guide future strategies in cancer management as they illustrate that alternating therapeutic pressure against distinct clonal populations is able to modulate clonal evolution. However, modulating EGFR-based genetic evolution with fixed therapeutic schedules alone may be insufficient to substantially alter clinical outcomes. Together they highlight the continuing requirement to develop improved monitoring strategies to not only track genetic evolution in real-time but to also understand non-genetic mechanisms of adaptation to therapeutic pressure.

## Methods
### Study design and objectives
Written informed consent was obtained from all participants. Ethics approval was obtained (Sydney Local Health District Ethics Review Committee (RPAH Zone)) and the study was conducted in accordance with the Declaration of Helsinki. The study design and conduct complied with all relevant regulations regarding the use of human study participants. An independent data and safety monitoring committee provided oversight of safety and efficacy. Study data was collected and managed by the NHMRC Clinical Trials Centre, Sydney, New South Wales. The study protocol is provided as a Supplementary Note in the Supplementary Information file.

OSCILLATE is a phase 2, single-arm, multicentre, open-label trial of alternating osimertinib with gefitinib in patients with advanced *EGFR* T790M positive NSCLC. Participants were recruited from 12 tertiary centres in Australia, from September 4, 2017 to June 11, 2019. The cut-off date for data analysis for this publication was December 11, 2021. The primary objective of the trial was PFS measured at the 12-month landmark. Secondary objectives include median PFS, time to first and second progression, feasibility of alternating treatment as defined by no dose interruptions for 6 months, ORR as defined by CR and PR, OS and adverse events. Exploratory objectives included analysis of changes in ctDNA and correlation with response to alternating third and first-generation EGFR TKI. Unplanned analysis included difference in PFS and OS between those with an *EGFR* exon 19 deletion compared to with exon 21 L858R, 12-month PFS rate by sex, and difference in PFS and OS between sex.

### Study population
Participants with histologically confirmed metastatic *EGFR* T790M mutation positive NSCLC and who had disease progression after first or second-generation EGFR-TKI were included. Eligibility criteria included documented evidence of *EGFR* T790M mutation on tissue and/or plasma analysis performed at either a central or local laboratory following disease progression on most recent EGFR-TKI therapy, ECOG performance status of 0 to 1, evaluable disease as defined by RECIST v1.1, life expectancy of >3 months, and adequate end-organ function. Exclusion criteria included previous or current treatment with osimertinib or other drugs that target *EGFR* T790M, uncontrolled brain metastases, and interstitial lung disease. For the purpose of this study, sex as a biological attribute was determined based on self-reporting. As there were no preferences on the selection of sex to be included, the study will result in a representative sex distribution, which should reflect the natural sex distribution of the underlying disease. There are no study findings that apply to only one sex. Based on previous observations, sex was not expected to affect survival of the trial participants. Unplanned analysis of the primary endpoint by sex was performed and reported in the results section. Participants were enrolled on a voluntary basis and were not compensated for participating on this study.

### Study treatment
Participants received an induction of continuous therapy with oral osimertinib 80 mg daily for 8 weeks. This was followed by alternating regimen of gefitinib 250 mg daily and osimertinib 80 mg daily in 4-week cycles (Fig. 1b). Treatment was continued until disease progression, unacceptable toxicity, death, or withdrawal of informed consent. Dose modifications according to the study protocol were permitted in the event of drug-related toxicity. Treatment with continuous dose osimertinib was allowed after progression on alternating therapy.

### Efficacy assessment
Tumor response was evaluated locally based on RECIST v1.1 by CT scan, which was performed at screening and every 8 weeks after starting study treatment until disease progression (Fig. 1b). The best overall response was defined as the best response recorded from the start of treatment until disease progression. Objective response was considered confirmed if the response was maintained at a subsequent scheduled CT assessment, at least 4 weeks after the criteria for response were first met.

### Blood collection and processing for ctDNA
Blood samples were collected in EDTA tubes for ctDNA analysis at screening (baseline), day 1 of each cycle (plus day 15 of cycles 3 and 4), and at the time of disease progression (Fig. 1b). Whole blood was first

centrifuged at 1600g for 10 min to separate the plasma from the peripheral blood cells, followed by a further centrifugation step at 20,000 g for 10 min to pellet any remaining cells and/or debris. The plasma was then stored at −80 °C until DNA extraction. DNA was extracted from 2 mls aliquots of plasma using the QIAmp Circulating Nucleic Acid Kit (Qiagen, Hilden, Germany) according to manufacturer's instructions. The DNA was eluted into 50μL buffed AVE (Qiagen) and stored at −20 °C.

### Targeted sequencing

Targeted capture-based sequencing of pre-treatment (baseline) and progression plasma samples was performed using the Avenio ctDNA analysis expanded kit (Roche diagnostics) following manufacturer's protocols (Supplementary Data 2 for list of genes). Based on the previously published cancer personalized profiling by deep sequencing (CAPP-Seq) methodology, this covers a panel of 77 genes optimized for use in CRC and NSCLC[34].

Between 6–10 ng of genomic DNA were used for library construction and the purified libraries were pooled. Libraries were quality-checked on an Agilent TapeStation. Sequencing was performed on an Illumina NextSeq 500 (150 bp paired end), with 8 samples per run (approximately 100 million PE reads/sample). Reads were aligned to the human genome (hg19) and data were analyzed using proprietary Roche Avenio oncology analysis software via a locally installed Roche server from a median sequencing depth of ~15,000X and a mean unique read depth of 3000X. The Roche analysis pipeline supports VAF detection of SNVs to 0.5% (50% sensitivity at detecting SNVs at 0.1%), targeted indels and fusions to 1%, and copy-number variations over 2.3-fold with sensitivities of >99%. Stated performance requires at least 60 million reads per sample which was achieved on all our samples[35]. To ensure sample integrity, longitudinal plasma samples were verified to come from the original participant by applying Somalier to targeted sequencing data[36].

### Digital PCR

Droplet digital PCR (ddPCR) analysis was performed using the Bio-Rad Droplet Digital PCR system following manufacturer's protocols. Allele-specific PCR assay to specifically detect and quantify the fractional abundance of the *EGFRm*, EGFR T790M mutation and corresponding wild-type allele was commercially obtained (BioRad Laboratories). For mutant-based assays, ddPCR reactions were 25 μL aqueous volumes that contained final concentrations of 1x ddPCR supermix for probes (without dUTP) (Bio-Rad), 0.9 μM each primer, 0.25 μM probe and between 0.05 to 5 ng of genomic DNA. The thermal cycling profile was 95 °C: for 10 min, followed by 55 cycles of 95 °C for 15 s and annealing for one minute at 55 °C. Each sample was analyzed by at least two technical replicates with 5μL DNA input per well. A Poisson correction was applied to determine the number of amplifiable molecules, which was used to further derive the number of copies of DNA carrying a particular mutation per millilitre of plasma. Data analysis was carried out using the QuantaSoft Software, version 1.7 (Bio-Rad). ctDNA was defined as detectable if there was ≥ 1 copy of mutant DNA detected in both duplicate reactions.

### Statistical Methods

A sample size of 45 participants was planned to distinguish the observed proportion alive and progression free at 12 months from 45% (not worthy to proceed to further evaluation) versus 65% (worthy to proceed) using a Simon's two-stage minimax design with 90% power and a 1-sided type 1 error rate of 10% (allowing for 4 non-evaluable participants). A futility analysis was planned (10/21 evaluable participants to be progression free at 12 months) however, recruitment completed before it could be conducted. Participant characteristics, treatment details and AEs were summarized using descriptive statistics. Safety analyses included all enrolled patients who fulfilled eligibility criteria and received one dose of the study treatment. Efficacy and biomarker analyses included all enrolled participants who fulfilled eligibility criteria, received at least one dose of the study treatment, and had at least one post-baseline efficacy assessment. Primary efficacy endpoint was PFS at 12 months. The response rates are estimated with 95% confidence interval calculated based on binomial distribution. Time-to-event endpoints (PFS, OS) are described using Kaplan-Meier methods with 95% confidence intervals. Cox proportional hazards regression was used to evaluate factors (exon 19 deletion, exon 21 L858R point mutation and sex) for association with PFS or OS. For biomarker analyses, group and sample comparisons were made using either two-tailed Mann-Whitney U test or Wilcoxon matched-pairs signed rank test, and Fisher's exact test was used to compare associations between categorical variables. All analyses were performed using R version 3.6.3 and GraphPad PRISM® version 9.1.2, where *P* values < 0.05 were considered significant.

### Reporting summary

Further information on research design is available in the Nature Portfolio Reporting Summary linked to this article.

## Data availability

The raw clinical data are protected and are not available due to data privacy laws. The de-identified datasets supporting the findings of this study are available for academic purposes on request from the corresponding authors, Professor Sarah-Jane Dawson (sarah-jane.dawson@petermac.org) and Professor Benjamin Solomon (ben.solomon@petermac.org) with the approval of the Institutional Ethics Committees for 5 years. The trial protocol is available as a Supplementary Note in the Supplementary Information. The sequencing dataset generated in this study is deposited under the following accession number in the European Genome-phenome Archive (EGA) (https://ega-archive.org/studies): EGAS50000000103. The raw sequencing data are available under controlled access due to privacy policy regulations and the data should only be used for research purposes only. Data are available upon request from corresponding author Professor Sarah-Jane Dawson for 5 years. All remaining data that support the findings of this study are available within the Article, the Supplementary Information or the Source Data file. Source data are provided in this paper. Source data are provided with this paper.

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

## Acknowledgements

The authors thank the patients and their families for participating in this study. OSCILLATE was supported by AstraZeneca, which provided osimertinib and gefitinib to support trial conducts. Additional support for translation research of the trial was supported by a National Health and Medical Research Council (NHMRC) grant (1158345, SJ-D, NP, BJS).

## Author contributions

Conception and design: Lavinia Tan, Sonia Yip, Nick Pavlakis, Martin R. Stockler, Sarah-Jane Dawson, Benjamin Solomon, Development of methodology: Lavinia Tan, Martin R. Stockler, Nick Pavlakis, Benjamin Solomon, Christopher Brown, Sarah-Jane Dawson, Acquisition of data (acquired and managed patients, provided facilities, etc.): Lavinia Tan, Antony Mersiades, Chee Khoon Lee, Sonia Yip, Thomas John, Steven Kao, Genni Newnham, Kenneth O'Byrne, Sagun Parakh, Victoria Bray, Kevin Jasas, Stephen Q. Wong, Sarah Ftouni, Jerick Guinto, Sushma Chandrashekar Martin R. Stockler, Stephen Clarke, Nick Pavlakis, Sarah-Jane Dawson, Benjamin Solomon, Analysis and interpretation of data (statistical analysis, biostatistics, computational analysis): Lavinia Tan, Chris Brown, Sarah-Jane Dawson, Benjamin Solomon, Administrative, technical, or material support (i.e., reporting or organizing data, constructing database): Lavinia Tan, Chris Brown, Antony Mersiades, Sarah Ftouni, Jerick Guinto, Sushma Chandrashekar, Writing, review, and/or revision of the manuscript: Lavinia Tan, Antony Mersiades, Chee Khoon Lee, Sonia Yip, Thomas John, Steven Kao, Genni Newnham, Kenneth O'Byrne, Sagun Parakh, Victoria Bray, Kevin Jasas, Stephen Q. Wong, Sarah Ftouni, Jerick Guinto, Sushma Chandrashekar Martin R. Stockler,

Stephen Clarke, Nick Pavlakis, Sarah-Jane Dawson, Benjamin Solomon, Study supervision: Benjamin Solomon, Sarah-Jane Dawson.

## Competing interests

L. Tan reports receiving speaker fees from Roche. C.K. Lee reports grants from AstraZeneca, Roche, Amgen, Merck KGA and honorarium from AstraZeneca, Amgen, GSK, Merck KGA, Novartis, Pfizer, Roche, and Takeda. He reports receiving travel fees from AstraZeneca. T. John reports receiving honorarium and is an advisory board member for BMS, AstraZeneca, Amgen, Roche, Pfizer, Takeda, Boehringer Ingelheim, MSD, Merck, Puma, Specialised Therapeutics, and Gilead. He reports receiving travel/speaker fees from AstraZeneca and MSD. S. Kao serves on advisory boards for AstraZeneca, Pfizer, Roche, BMS, MSD, and Takeda. He reports receiving honorarium to institution for MSD, BMS, Roche, AstraZeneca, Pfizer, Boeringher Ingelheim and travel support from BMS, Roche, AstraZeneca, and Boeringher Ingelheim. K. O'Byrne reports receiving honorarium from Merck Sharp & Dohme, BMS, Roche, Boehringer Ingelheim, AstraZeneca, Pfizer/EMD Serano, Novartis, Janssen-Cilag, Yuhan, Tristar Technology Group, Takeda, Amgen, BeiGene. He has consulting/advisory role for Merck Sharp & Dohme, Boehringer Ingelheim, Roche/Genentech, Janssen-Cilag, Pfizer, AstraZeneca/MedImmune, BMS, Novartis, Yuhan, Sanofi, Amgen, BeiGene. He has stock and other ownership interests with Carpe Vitae Pharmaceuticals, Replica Pharmaceuticals, DGC diagnostics and has four active patents (two published and two provisional). He also has speaking roles for Merck Sharp & Dohme, Boehringer Ingelheim, BMS, Roche, Janssen. S. Parakh reports receiving research funding from Bayer and Roche, honorarium from BMS and Roche. He is an advisory board member for AstraZeneca and MSD. S. Q. Wong reports receiving speaker fees from Roche. S. Clarke is an advisory board member for AstraZeneca and Beigene and has speaking roles for AstraZeneca. N. Pavlakis reports consulting or advisory role for Roche, Boehringer Ingelheim, AstraZeneca, Merck KGA, Merck Serono, Amgen, Merck Sharp & Dohme, Novartis, Pfizer, Takeda, and BMS. M.R. Stockler reports receiving research funding from Astellas, Celgene, Bayer, Bionomics, Medivation, Sanofi, Pfizer, AstraZeneca, BMS, Roche, Amgen, Merck Sharp & Dohme, Tilray, and BeiGene. He has speaking roles for Astellas Pharma and Medivation/Pfizer. SJ. Dawson has received research funding from Cancer Therapeutics CRC (CTx) and Genentech. She has been an advisory board member for AstraZeneca,

Inivata and Adela. B. J. Solomon reports receiving honorarium and is an advisory board member for AstraZeneca, Pfizer, Novartis, Roche, Eli Lilly, Amgen, Merck, Bristol Myers Squibb, Takeda, Janssen, and BeiGene. No potential conflicts of interest were disclosed by the other authors.

## Additional information

¹Peter MacCallum Cancer Centre, Melbourne, Vic, Australia. ²Sir Peter MacCallum Department of Oncology, The University of Melbourne, Melbourne, VIC, Australia. ³NHMRC Clinical Trials Centre, University of Sydney, Sydney, NSW, Australia. ⁴St George Hospital, Sydney, NSW, Australia. ⁵Chris O'Brien Lifehouse, Sydney, NSW, Australia. ⁶St Vincent's Hospital, Melbourne, VIC, Australia. ⁷Princess Alexandra Hospital, Brisbane, QLD, Australia. ⁸Austin Hospital, Olivia Newton John Cancer and Wellness and Research Centre, Melbourne, VIC, Australia. ⁹Liverpool Hospital, Sydney, NSW, Australia. ¹⁰Sir Charles Gairdner Hospital, Perth, WA, Australia. ¹¹Royal North Shore Hospital, Sydney, NSW, Australia. ¹²University of Sydney, Sydney, NSW, Australia. ¹³Centre for Cancer Research, The University of Melbourne, Melbourne, VIC, Australia. ¹⁴These authors jointly supervised this work: Sarah-Jane Dawson, Benjamin J. Solomon. ✉e-mail: Sarah-Jane.dawson@petermac.org; ben.solomon@petermac.org

