## [Peer Review File · Nature Communications]

Alternating osimertinib and gefitinib therapy in advanced EGFR-T790M positive non-small cell lung cancer: the OSCILLATE trialReviewers' Comments:

Reviewer #1:

Remarks to the Author:

The authors have answered my questions that I had previously, and I thank them for now providing adequate details in the bioinformatics methods.

Related to the clarification from the authors that Figure 2C and 2D are using different timescales (Figure 2D is only showing responses after 9 months), then I ask why Figure 2D is not showing the entire timescale or at least that this is mentioned in the legend.

Regarding sample swaps, the authors state "We have experience using Sommelier to ensure sample integrity, and all care was taken during our sequencing workflow to ensure that samples were not swapped.". Can the authors please confirm that Sommelier has been used to verify sample identity on these samples, because "experience using" is a vague term that does not confirm whether it has actually been used here. I would encourage the addition of a sentence in the methods to confirm this to, and I would highly encourage the use of such tools in any study of this kind.

I have no further comments on the manuscript apart from these.

Reviewer #2:

Remarks to the Author:

The authors have addressed my prior comments.

My only additional comments relate to the supplemental data, which should be expanded to allow readers to reproduce their analyses. This includes adding a table that contains key patient level data such as number of cycles of osimertinib and gefitinib, best overall response, progression free survival time/status, and overall survival time/status. Additionally, they should include a table that lists the volume of plasma used for cfDNA isolation and the cfDNA concentration (ng/ml) for each sample.

Reviewer #4:

Remarks to the Author:

This manuscript presents the results from OSCILLATE trial -- evaluating the treatment effects of alternating osimertinib and gefitinib therapy modulates distinct clonal dynamics in 2 advanced EGFR-T790M positive non-small cell lung cancer. I included my comments below for authors' considerations.

1. In oncology, "predictive" and "prognostic" refer to two different concepts. Predictive values refer to the biomarker status indicates different treatment effect on endpoints which involves comparisons of endpoint between experimental vs. control treatment within the biomarker subgroups, separately. A significant interaction effect (biomarker status and treatment assignment) on endpoint indicates predictive value. Prognostic value refers to a significant association between the biomarker status and clinical endpoints. In this manuscript, the analysis regarding ctDNA seems a prognostic analysis, not predictive analysis. It would be recommended to use the term of "prognostic" rather than "predictive" throughout the manuscript.

2. In the abstract, the statement of "Importantly, the alternating regimen was associated with frequent loss of T790M ctDNA in 73% of participants and no evidence of the EGFR C797S resistance mutation at disease progression." is not accurate. There is no comparison to a "control" treatment (ideally randomized), so this is merely an observation, not a proven association. It would be more

appropriate if it is stated as: "Importantly, among patients treated with the alternating regimen, there were 73% of participants with frequent loss of T790M ctDNA and no evidence of the EGFR C797S resistance mutation at disease progression." The similar edits would be recommended in the manuscript text too.

3. From the protocol, the trial design is a single arm phase II trial with a Simon's 2-stage, minimax design. The null hypothesis is 12m PFS \leq 45% and the alternative hypothesis is 12m PFS \geq 65%. The targeted sample size is 41 evaluable (inflated to 45 for unevaluable pts) patients to provide 90% power at one-sided alpha of 0.1. In "statistical methods" section, it was stated "A sample size of 45 participants..." This is not accurately presenting the statistical design of the trial. Also, the pre-planned interim analysis should be included in the methods section, as well as, the interim analysis timing and results should be reported in the results section.

4. The trial efficacy hypothesis testing was based on a binary endpoint of PFS at 12m with pre-specified cutoffs (stated in protocol but not in manu): at interim – claim "fail to show efficacy" if \leq 9 pts of first 21 evaluable patients are alive and progression free at 12m; at final analysis – claim "warrant further evaluation", if \geq 23 of the first 41 evaluable patients are alive and progression free at 12m. The data in manuscript should be rephrased according to the original statistical design. The correct hypothesis testing is not comparing the 12m PFS rate based on KM to 65% (the alternative target rate). The presentation of the results regarding hypothesis testing portion needs to be reviewed and revised by a statistician. With such low 12 PFS rate (38%), I am wondering if the trial should've stopped its accrual after interim analysis?

5. The objectives regarding ctDNA were tertiary in trial protocol. As expected, the analyses were not powered and were in exploratory nature. The findings should be interpreted as hypothesis generating and with extra cautions. The current manuscript placed a lot of emphasis on ctDNA related results which seems over-interpreting the results. Furthermore, the sample size for ctDNA dynamics analyses is reduced substantially due to availability of the samples at post-baseline timepoints. Conducting pairwise (timepoints) comparisons rather than longitudinal analyses inflated type I error rate further. Thus, without an independent validation cohort, the current results regarding ctDNA should be deemphasized in this manuscript.

Reviewer #5:

Remarks to the Author:

These authors evaluated the efficacy of alternative EGFR inhibitors osimertinib and gefitinib treatment in 47 patients with lung cancer that acquired resistance to prior first- or second-generation EGFR inhibitors as a single arm phase 2 trial. Additionally, they evaluated baseline and acquired resistance mechanisms by serial circulating tumor DNA (ctDNA) analyses. The primary endpoint of 12-month PFS of 65% was not met.

Major comments

1) Given the unsuccessful achievement of primary endpoint and the out-of-date regimen, the most important data of this paper is no detection of EGFR C797S mutation during alternative treatment. Therefore, these authors need to precisely describe cases related to C797S even if the number of these cases are as small as two. Importantly, C797S was not detected in any cases during alternative treatment but subsequent osimertinib only induced C797S in 2 patients. Please clearly describe these data in Abstract and change the discussion line 486-487 "delayed the emergence of C797S mediated resistance in all cases" which overstate.

2) One of the big reasons why this trial did not meet the primary endpoint would be the inclusion of MET-amplified cohort in this trial. It is widely established that MET amplification is a mechanism of acquired resistance to EGFR inhibitors and combination with MET inhibitor is essential to overcome.

Scientifically, these patients do not make sense and mask the efficacy of alternative treatment because both osimertinib and gefitinib are resistant to them. Subset analyses of RR, 12-month PFS without MET-amplified cohort would be helpful to indirectly assess the real efficacy of the alternative treatment.

3) The strength of this study is that ctDNA samples were analyzed not only at the timing of progression to alternative treatment (Fig. 4f. here no C797S cases) but also subsequent resistance to osimertinib monotherapy beyond PD (Suppl Fig 20). Although the patient cohort of these figures are a little different, this reviewer strongly recommends that these authors try to combine these 2 figures into one so that the readers can easily follow the change of resistance mechanisms over time with emphasizing on C797S cases. Please explain why the important patient ID 29 is not listed in Fig 4f.

4) Please annotate acquired genetic alterations show in Fig. 4f and Suppl Fig 20 based on previous evidence. In terms of both clinical and scientific viewpoints, it's important to differentiate resistance-causing alterations and merely passenger alterations. For example, "EGFR amplification" should be clarified which allele (mutant or wildtype) were amplified because only wildtype amplification can show resistance to mutant-selective osimertinib. Please try to annotate all other mutations as well at least such as minor EGFR S492R/R451C mutations.

5) Please add the explanation/interpretation about the data showing increase in W12 compared with W4.

Minor comments

1) Supp Fig 6b: Median OS of "15mo" and "26mo" are reversed in the figure.

2) P.12 line 307: Supplementary Fig. "7" should be "8". Similarly, p.12 line 317 Supplementary Fig "11" should be "10". Please extensively check all supp figure numbers throughout the paper because this type of mistakes could significantly reduce the reliability of entire data in this paper.

3) In discussion p.15 line 403 "additional median PFS2 of 5 months" should be described in result section as well because this is an important data which is not described.

REVIEWER COMMENTS

Reviewer #1 (Remarks to the Author):

The authors have answered my questions that I had previously, and I thank them for now providing adequate details in the bioinformatics methods.

Related to the clarification from the authors that Figure 2C and 2D are using different timescales (Figure 2D is only showing responses after 9 months), then I ask why Figure 2D is not showing the entire timescale or at least that this is mentioned in the legend.

We thank the reviewer for this suggestion and have now included a new Figure 2d with a timescale of 24 months. The patient who achieved a complete response at 18 months is now represented in this new figure.

Regarding sample swaps, the authors state "We have experience using Sommeliere to ensure sample integrity, and all care was taken during our sequencing workflow to ensure that samples were not swapped.". Can the authors please confirm that Sommeliere has been used to verify sample identity on these samples, because "experience using" is a vague term that does not confirm whether it has actually been used here. I would encourage the addition of a sentence in the methods to confirm this to, and I would highly encourage the use of such tools in any study of this kind.

We can confirm that Sommeliere has been used to verify sample identity and there were no sample swaps. We have added this in the methods section.

I have no further comments on the manuscript apart from these.

Reviewer #2 (Remarks to the Author):

The authors have addressed my prior comments.

My only additional comments relate to the supplemental data, which should be expanded to allow readers to reproduce their analyses. This includes adding a table that contains key patient level data such as number of cycles of osimertinib and gefitinib, best overall response, progression free survival time/status, and overall survival time/status. Additionally, they should include a table that lists the volume of plasma used for cfDNA isolation and the cfDNA concentration (ng/ml) for each sample.

We thank the reviewer for this suggestion. Individual patient clinical data will be made available on request. We have included a table (Supplementary Table 3) that lists the volume of plasma used for cfDNA isolation and the cfDNA concentration.

Reviewer #4 (Remarks to the Author):

This manuscript presents the results from OSCILLATE trial -- evaluating the treatment effects of alternating osimertinib and gefitinib therapy modulates distinct clonal dynamics in 2 advanced EGFR-T790M positive non-small cell lung cancer. I included my comments below for authors' considerations.

1. In oncology, "predictive" and "prognostic" refer to two different concepts. Predictive values refer to the biomarker status indicates different treatment effect on endpoints which involves comparisons of endpoint between experimental vs. control treatment within the biomarker subgroups, separately. A significant interaction effect (biomarker status and treatment assignment) on endpoint indicates predictive value. Prognostic value refers to a significant association between the biomarker status and clinical endpoints. In this manuscript, the analysis regarding ctDNA seems a prognostic analysis, not predictive analysis. It would be recommended to use the term of "prognostic" rather than "predictive" throughout the manuscript.

We thank the reviewer for this recommendation and have now used the term prognostic in the manuscript.

2. In the abstract, the statement of "Importantly, the alternating regimen was associated with frequent loss of T790M ctDNA in 73% of participants and no evidence of the EGFR C797S resistance mutation at disease progression." is not accurate. There is no comparison to a "control" treatment (ideally randomized), so this is merely an observation, not a proven association. It would be more appropriate if it is stated as: "Importantly, among patients treated with the alternating regimen, there were 73% of participants with frequent loss of T790M ctDNA and no evidence of the EGFR C797S resistance mutation at disease progression." The similar edits would be recommended in the manuscript text too.

We thank the reviewer for this recommendation and have now rephrased this in the abstract and discussion.

3. From the protocol, the trial design is a single arm phase II trial with a Simon's 2-stage, minimax design. The null hypothesis is 12m PFS \leq 45% and the alternative hypothesis is 12m PFS \geq 65%. The targeted sample size is 41 evaluable (inflated to 45 for unevaluable pts) patients to provide 90% power at one-sided alpha of 0.1. In "statistical methods" section, it was stated "A sample size of 45 participants..." This is not accurately presenting the statistical design of the trial. Also, the pre-planned interim analysis should be included in the methods section, as well as, the interim analysis timing and results should be reported in the results section.

We thank the reviewer for pointing this out and have now amended the text in the statistical analysis section of the main manuscript. A futility analysis was planned (10/21 evaluable participants to be progression free at 12 months). Study recruitment closed before the data was available to evaluate this, thus it was not conducted.

4. The trial efficacy hypothesis testing was based on a binary endpoint of PFS at 12m with pre-specified cutoffs (stated in protocol but not in manu): at interim – claim "fail to show efficacy" if \leq 9 pts of first 21 evaluable patients are alive and progression free at 12m; at

final analysis – claim “warrant further evaluation”, if ≥ 23 of the first 41 evaluable patients are alive and progression free at 12m. The data in manuscript should be rephrased according to the original statistical design. The correct hypothesis testing is not comparing the 12m PFS rate based on KM to 65% (the alternative target rate). The presentation of the results regarding hypothesis testing portion needs to be reviewed and revised by a statistician. With such low 12 PFS rate (38%), I am wondering if the trial should've stopped its accrual after interim analysis?

The text has been reworded, explicitly referring to the test of the null hypothesis was not rejected. *“The trial, however, did not reach the pre-specified target required to warrant further evaluation (23/41 participants progression-free at 12 months).”* and *“This was below the pre-specified target of 23/41 progression free at 12-months.”*

5. The objectives regarding ctDNA were tertiary in trial protocol. As expected, the analyses were not powered and were in exploratory nature. The findings should be interpreted as hypothesis generating and with extra cautions. The current manuscript placed a lot of emphasis on ctDNA related results which seems over-interpreting the results. Furthermore, the sample size for ctDNA dynamics analyses is reduced substantially due to availability of the samples at post-baseline timepoints. Conducting pairwise (timepoints) comparisons rather than longitudinal analyses inflated type I error rate further. Thus, without an independent validation cohort, the current results regarding ctDNA should be deemphasized in this manuscript.

We are aware of the limitations of the study and agree that results of the biomarker analyses are exploratory and would require validation in an independent cohort to support our findings. We have further emphasised this in the discussion.

Reviewer #5 (Remarks to the Author):

These authors evaluated the efficacy of alternative EGFR inhibitors osimertinib and gefitinib treatment in 47 patients with lung cancer that acquired resistance to prior first- or second-generation EGFR inhibitors as a single arm phase 2 trial. Additionally, they evaluated baseline and acquired resistance mechanisms by serial circulating tumor DNA (ctDNA) analyses. The primary endpoint of 12-month PFS of 65% was not met.

Major comments

1) Given the unsuccessful achievement of primary endpoint and the out-of-date regimen, the most important data of this paper is no detection of EGFR C797S mutation during alternative treatment. Therefore, these authors need to precisely describe cases related to C797S even if the number of these cases are as small as two. Importantly, C797S was not detected in any cases during alternative treatment but subsequent osimertinib only induced C797S in 2 patients. Please clearly describe these data in Abstract and change the discussion line 486-487 “delayed the emergence of C797S mediated resistance in all cases” which overstate.

We thank the reviewer for this suggestion and have now described the 2 cases of EGFR C797S resistance mutation detected in plasma post continuous osimertinib in the manuscript. Both participants harboured an EGFR exon 19 deletion mutation, however, only

OSC014 had detectable T790M on plasma testing at baseline. Of note, T790M mutation was maintained at both progression time points for OSC014, with acquired *MET* amplification and *TP53* mutation post alternating and continuous osimertinib. OSC029 had a detectable *FGFR2* mutation post alternating therapy and no other detectable genomic mutation post continuous osimertinib. The statement in line 486-487 has also been modified.

2) One of the big reasons why this trial did not meet the primary endpoint would be the inclusion of MET-amplified cohort in this trial. It is widely established that MET amplification is a mechanism of acquired resistance to EGFR inhibitors and combination with MET inhibitor is essential to overcome. Scientifically, these patients do not make sense and mask the efficacy of alternative treatment because both osimertinib and gefitinib are resistant to them. Subset analyses of RR, 12-month PFS without MET-amplified cohort would be helpful to indirectly assess the real efficacy of the alternative treatment.

We thank the reviewer for this suggestion and have conducted this post-hoc analysis. Within 36 non-*MET*-amplified patients, RR was higher 18/36 (56%) vs 2/11 (18%), PFS was longer 11m vs 3.7m (HR=2.17, 95% CI 1.06-4.55) and OS was longer 38m vs 15m (HR=1.82, 95% CI 0.76-4.35).

3) The strength of this study is that ctDNA samples were analyzed not only at the timing of progression to alternative treatment (Fig. 4f. here no C797S cases) but also subsequent resistance to osimertinib monotherapy beyond PD (Suppl Fig 20). Although the patient cohort of these figures are a little different, this reviewer strongly recommends that these authors try to combine these 2 figures into one so that the readers can easily follow the change of resistance mechanisms over time with emphasizing on C797S cases. Please explain why the important patient ID 29 is not listed in Fig 4f.

We thank the reviewer for this suggestion and have now combined figure 4f and Supplementary Fig. 20.

4) Please annotate acquired genetic alterations show in Fig. 4f and Suppl Fig 20 based on previous evidence. In terms of both clinical and scientific viewpoints, it's important to differentiate resistance-causing alterations and merely passenger alterations. For example, "EGFR amplification" should be clarified which allele (mutant or wildtype) were amplified because only wildtype amplification can show resistance to mutant-selective osimertinib. Please try to annotate all other mutations as well at least such as minor EGFR S492R/R451C mutations.

We thank the reviewer for this suggestion. All mutations are detailed in supplementary table 3 along with COSMIC ID if reported.

5) Please add the explanation/interpretation about the data showing increase in W12 compared with W4.

The increase in W12 compared to W4 ctDNA levels were seen post alternating gefitinib and osimertinib treatment, in some patients where the EGFRm levels steadily increased to pre-treatment levels. The increase in W12 compared to W4 ctDNA levels were seen in patients who had progressive disease in response to alternating treatment.

Minor comments

1) Supp Fig 6b: Median OS of “15mo” and “26mo” are reversed in the figure.

We thank the reviewer for pointing this out and the figure has now been corrected.

2) P.12 line 307: Supplementary Fig. “7” should be “8”. Similarly, p.12 line 317

Supplementary Fig “11” should be “10”. Please extensively check all supp figure numbers throughout the paper because this type of mistakes could significantly reduce the reliability of entire data in this paper.

We have now corrected and checked all supplementary figure numbers in the manuscript.

3) In discussion p.15 line 403 “additional median PFS2 of 5 months” should be described in result section as well because this is an important data which is not described.

The median PFS is now described in the results section.

Reviewers' Comments:

Reviewer #4:

Remarks to the Author:

I appreciate authors' responses and the edits to the manuscript improved it from statistical point of view. I do not have further comments.

Reviewer #5:

Remarks to the Author:

These authors addressed all of this reviewer's scientific comments. This reviewer believes that this manuscript was greatly improved.